# Targeting a Subset of the Membrane Proteome for Top–Down Mass Spectrometry: Introducing the Proteolipidome

**DOI:** 10.3390/proteomes8010005

**Published:** 2020-03-10

**Authors:** Julian Whitelegge

**Affiliations:** The Pasarow Mass Spectrometry Laboratory, Jane and Terry Semel Institute of Neuroscience and Human Behavior, David Geffen School of Medicine, University of California Los Angeles, Los Angeles, CA 90095, USA; jpw@chem.ucla.edu; Tel.: +1-310-794-5156

**Keywords:** proteoform, FTICR, CAD, aiECD, high-resolution mass spectrometry

## Abstract

A subsection of integral membrane proteins partition into chloroform during a chloroform/methanol/water extraction primarily designed to extract lipids. Traditionally, these proteins were called proteolipids due to their lipid-like properties; the c-subunit of the ATP synthase integral FO component is the best known due to its abundance. In this manuscript, we investigate purification of proteolipid proteins away from lipids for high-resolution mass spectrometry. Size-exclusion chromatography on silica beads using a chloroform/methanol/aqueous formic acid (4/4/1; *v*/*v*) mobile phase allowed the separation of larger proteins (>3 kDa) from lipids (<1.5 kDa) and analysis by online electrospray ionization mass spectrometry. Fraction collection for mass spectrometry was limited by presence of plasticizers and other contaminants solubilized by chloroform. Drying down of the protein sample followed by resuspension in formic acid (70%) allowed reverse-phase chromatography on a polymeric support at elevated temperature, as described previously. Fractions collected in this way could be stored for extended periods at −80 °C without adducts or contaminants. Top–down mass spectrometry enabled the definition of PsaI as a novel proteolipid of spinach thylakoid membrane. Proteolipid preparation worked similarly when total membranes from mouse brains were extracted with chloroform. While it might be tempting to use the described extraction, we prefer to broaden the meaning of the term, whereby the *proteolipidome* is defined as a novel biological membrane proteome that includes the full complement of membrane proteins, their binding partners/ligands and their tightly bound structural lipids that constitute each protein–lipid complex’s functional unit; that is, a complete description of a biological membrane.

## 1. Introduction

Biological membranes are critical to cell function and survival, providing a selectively permeable barrier that allows development of gradients of ions and charge and other molecules. While a fluid bilayer of lipids is characteristic of nearly all membranes, their unique properties are also determined by the proteins that constitute a highly significant proportion of their mass. Singer and Nicholson [1] divided the lipids into the 95% that make up the general bilayer distinct from 5% that are bound tightly to proteins and can be thought of as *structural lipids*. Membrane proteins are made up of integral components that span the bilayer and peripheral ones that associate with the membrane surface either by interaction with lipids or integral proteins or both. A complete description of a biological membrane should include detail of intimate protein–lipid interactions and any other molecular factors that modulate function under native conditions including small-molecule ligands. The majority of drugs act upon membrane protein–lipid complexes.

A definition of the membrane proteome has been addressed through a wide variety of techniques that render proteins or parts of them amenable to mass spectrometry. Membranes have been solubilized with sodium dodecyl sulfate (SDS) prior to electrophoresis, ‘in-gel’ digestion with trypsin and extraction of soluble peptides from the gel [2]. Alternatively, many involved in shotgun proteomics have used solubilization in different detergents or organic solvents to yield some useful peptides for mass spectrometry [3]. Alternative enzymes or chemical reagents have been used to improve peptide recovery [4] and chromatography has been modified to improve yield of transmembrane regions of the membrane [5]. Since it targets hydrophobic residues, chymotrypsin is more suitable (or complementary to) than trypsin for membrane proteins [6,7,8]. Intact membrane proteins can be subjected to chromatography under specialized conditions facilitating top–down mass spectrometry that is inclusive of the transmembrane domains [9,10]. Most recently, breakthroughs in native mass spectrometry have allowed intact analysis of membrane protein complexes with bound lipids and co-factors, providing a fascinating picture of components of the biological membrane [11]. A novel protocol that gives access to the membrane proteome via an unusual property is described herein.

It has been known for many years that some proteins partition into chloroform during lipid extraction and these were called proteolipids in recognition of this lipid-like property [12]. Perhaps the best known of the proteolipids is the c-subunit of the ATPase-synthase FO due to its very high relative abundance [13], and whose structure was determined by NMR in a 4/4/1 solvent mixture [14]. This protein, isolated from the chloroplast thylakoid membrane, was the first integral membrane protein to be analyzed by top–down mass spectrometry [15] using a size-exclusion chromatography (SEC) procedure that largely separates proteolipids from lipids [9,16,17]. Other proteins were more elusive due to the complex mixture that elutes somewhat simultaneously due to the relatively poor resolution of SEC. In this paper, SEC is used to purify proteolipids before further separation using reverse-phase chromatography on a polymeric stationary phase at elevated temperature, as reported previously [6,9,18]. Proteolipids isolated in this way were amenable to top–down mass spectrometry with both collisionally activated and electron capture dissociation (CAD and ECD), allowing proteome-wide identification in a database search. Proteolipids should not be confused with lipoproteins; proteolipids are proteins with lipid-like properties that partition into chloroform in an aqueous/organic phase separation, while lipoproteins are soluble proteins (not membrane proteins) that bind lipids, rendering them soluble as part of a lipid/protein complex, for example, Apo A–I and the high-density lipoprotein particle (HDL). Covalent lipidation of a membrane protein increases its hydrophobicity, though we have yet to find an example where it causes it to partition into chloroform.

## 2. Materials and Methods

### 2.1. Materials

Spinach was obtained fresh from a local market and thylakoid membranes were prepared by homogenization and centrifugation. Briefly, 200 gm leaf was homogenized in 500 mL ice cold buffer in a large Waring blender (3 × 30 s full power on /30 s power off) as previously described [6]. Yield of thylakoids was 250 mg chlorophyll and aliquots were stored at −80 °C. Mouse brains from male C57BL6 mice were dissected post sacrifice and stored at −80 °C. Brain sample (0.5 gm) was homogenized in 1 mL of PBS (Phosphate Buffered Saline has pH = 7.4; 137 mM NaCl, 2.7 mM KCl, 10 mM Na_2_HPO_4_, 1.8 mM K_2_HPO_4_) for 2 min in a 2 mL Teflon/glass homogenizer, clarified by centrifugation at low speed (2000 rpm; 5 min) and the supernatant extracted with chloroform. All manipulations were performed at 4 °C, until chloroform extraction that was performed at 24 °C.

### 2.2. Methods

Chloroform extraction was performed using the procedure of Wessel and Flugge [19]; see Figure 1. Aqueous sample (50 µL) was diluted with water (200 µL), and methanol (600 µL) and then chloroform (200 µL) were added prior to mixing. Water (400 µL) was added, inducing phase separation, and the sample was mixed vigorously (2 min). The sample was centrifuged (13,000× g) for 5 min and the lower phase was removed using a 0.5 mL Hamilton syringe. This lower chloroform-containing phase was stored at 4 °C for no more than 24 h. Aliquots (100 µL) were removed for HPLC.

Size-exclusion chromatography (SW 2000 XL; 4.6 × 300 mm; Tosoh Biosciences) in chloroform:methanol:1% aqueous formic acid (4/4/1; *v*/*v*; 40 °C; 250 µL/minute) was performed isocratically as described previously [16]. Aliquots (100 µL) of chloroform extract were injected via a 100 µL loop onto a column previously equilibrated in the same buffer. Column eluent was directed to the source of an electrospray ionization mass spectrometer operated in the positive ion mode and a liquid flow splitter (50%) allowed collection of fractions every minute (125 µL fractions) through elution of proteolipids (fractions 6 and 7). Alternatively, the entire elution was diverted for this period in order to maximize yield of proteolipids (Figure 1). Fractions containing lipids (8 min onwards) were diverted from the mass spectrometer and discarded. A syringe pump HPLC (Applied Biosystems 140b) was used for isocratic size-exclusion chromatography.

Proteolipid fractions at 6–8 min were dried down and combined with the same fractions from 4 other identical runs, dissolved in 100 µL 70% formic acid and immediately injected on to a pre-equilibrated reverse-phase column. Reverse-phase chromatography on a polymeric column (PLRP/S; 300 Å × 5 µm; 2 × 150 mm; Agilent Technologies) at elevated temperature (40 °C) was performed as previously described [9]. Solutions were 0.1% trifluoroacetic acid (TFA) in water (A) and a 50/50 mixture of 0.1% TFA in acetonitrile with isopropanol (B). The column was equilibrated in 95% A and 5% B for 20 min at a flow rate of 120 µL/minute prior to gradient elution initiating 5 min after injection. A linear gradient runs from 5% B at 5 min to 60% B at 40 min and then to 99% B at 55 min. A quaternary pump HPLC (Agilent 1200) was used for gradient reverse-phase chromatography.

Online (Figure 1) low-resolution electrospray ionization mass spectrometry directly on column effluent was performed on an ion trap instrument (LTQ; Thermo Fisher) using an Ion Max source operated in the positive ion mode (4000 V) with standard parameters (capillary temperature 275 °C; capillary voltage 50 V; tube lens 200 V). Mass spectra were acquired from 600 to 2000 *m/z* with 4 microscans.

Offline (Figure 1) top–down high-resolution mass spectrometry was performed on fractions collected during LC–MS+ using a hybrid ion-trap 7 Tesla Fourier-transform ion cyclotron resonance mass spectrometer (LTQ FT Ultra; Thermo Fisher) using static nano-electrospray (nanospray source; Thermo Fisher). Collisionally activated dissociation (CAD) was accomplished in the ion trap (He collision gas) using ion energy sufficient to dissociate approximately 90% precursor, typically 12–15 normalized collision energy (the other settings were as standard). Activated-ion electron capture dissociation (aiECD) was accomplished within the ICR cell using the IR laser to activate the ions and the electron filament to produce electrons as described [15]. The laser was operated at 40% maximum intensity for 400 ms in parallel with electron capture at 1.5 eV electron energy for 400 ms. Typically, 100 transients were averaged at a resolution of 100,000 at 400 *m*/*z*.

Informatics data was obtained using Qualbrowser (Thermo) software to review spectra, MagTran software to deconvolute low-resolution mass spectra and ProSight 4.1 softwareto process high-resolution data. MagTran was operated with a mass range of 3000–30,000 and with a charge range of 2–30. Prosight was operated in the ‘sequence tag’ mode for protein identification using sequence tags and the ‘absolute mass’ mode once the protein was identified. Signal/noise was set at 7 and the filter that limits the number of ions identified was deactivated. The Prosight database for spinach includes 23,435 entries for the nuclear genomic sequence plus 91 entries for the chloroplast plastid genome (https://www.uniprot.org/proteomes/UP000054095; 5th November 2019).

## 3. Results

Lipid extractions into chloroform from biological membrane systems were described by Folch [20] and Bligh and Dyer [21]. A variant is the chloroform/methanol/water protocol used as described by Wessel and Flugge [19] specifically to precipitate proteins away from detergents. Many membrane proteins are precipitated along with soluble proteins at the aqueous/organic interface. The lower phase was collected with a stainless steel/glass (Hamilton-type) syringe and stored at 4 °C for a maximum of 24 h prior to further chromatography (the lipid/proteolipid mixture is susceptible to oxidation and plasticizer contamination if stored longer). Aliquots (100 µL) were injected onto a size-exclusion column as described (see Figure 1). The proteolipid peak was diverted to collect these proteins and the later eluting lipids were diverted to waste (see base peak chromatogram shown for SEC) (Figure 1). Proteins that precipitate in the Wessel and Flugge procedure (see Figure 1) can be dissolved in formic acid and crudely separated by the same size-exclusion chromatography (SEC) method [9,16]. Figure 2A shows a 3D ion map chromatogram that emphasizes elution of proteins (6–8 min) versus lipids (>8 min). Figure 2B shows the mass spectrum of the proteolipids. Deconvolution of the multiple signals of unclear charge performed quite poorly (not shown), emphasizing the need to further separate these species or use high-resolution mass spectrometry to reveal charge state. The strongest signals at *m*/*z* 1334.5 and 1601.17 correspond to the 6- and 5-charge states of the ATP synthase c-subunit of molecular weight 8002 Da that consists of residues 1–81, with retention of the initiating formyl methionine residue (calculated average mass 8002.4 Da), as we previously described [15].

Past experiments were hampered by the need to rapidly analyze the protein eluent, as extended storage in microcentrifuge tubes lead to plasticizer contamination and storage in glass tubes resulted in Na^+^ and other contaminant ion adducts [15]. Here, it is reported that the protein fraction can be immediately dried by vacuum centrifugation and resuspended in 70% formic acid for injection to reverse-phase chromatography using a polymeric stationary phase at elevated temperature [6,9,18]. Fractions collected into microcentrifuge tubes can subsequently be stored at −80 °C for extended periods and shipped on dry ice if necessary. Figure 3A shows a typical separation achieved when thylakoid membrane proteolipids are separated by reverse-phase chromatography after substantial removal of lipids by SEC. A peak of *m*/*z* 1233 corresponding to a mass of 3696.9 Da was selected for tandem mass spectrometry to show that this method could be used to identify the protein in a large database. Figure 3B inset shows the ion isolation used, providing the charge state and thus the unequivocal identification of the molecular mass of the protein. Figure 3B,C demonstrates the results of static nanospray experiments on fraction 29 (elution 29.5 min) to identify and characterize a small protein of mass 3696.9 Da using high-resolution Fourier-transform ion cyclotron resonance (FT-ICR) mass spectrometry with both collisionally activated dissociation (CAD) and activated-ion electron capture dissociation (aiECD) [15]. Sequence tags were obtained from both the CAD spectrum (Figure 3B) and the aiECD spectrum (Figure 3C) allowing identification of the PsaI protein from the *S. oleracea* database. Protein identification can be a challenge because transmembrane domains use a limited repertoire of amino acid residues, giving higher conservation than for a general protein sequence. It is not yet clear whether CAD or aiECD will be more useful for protein identification. Nevertheless, both dissociation mechanisms give complementary information. This protein retained its initiating formyl methionine residue accounting for the extra 28 Da mass, as is typically observed for many integral membrane proteins translated in organelles [9,22,23]. This is due to the prokaryotic origin of these organelles. The *N*-terminal situation of the formyl group is proven by the fact that all b- and c-ions, and no y- and z-ions, include this modification. Retention of the initiating formyl methionine should be distinguished from the artefactual formylation of internal residues at high concentrations of formic acid, which we effectively eliminate by limiting formic acid exposure time to less than 2 min [24]. A small amount of a protein of mass 3712.8 Da was observed to elute ~1 min prior to the dominant 3696.9 Da, hypothetically due to oxidation of methionine residues. It is assumed that all the multiply charged ions seen in the SEC separation (Figure 2) result from proteolipids but until we have formally proven this to be so, we have to entertain the possibility that some other biological macromolecule might be included in this fraction.

To demonstrate applicability to proteins of biomedical importance, the proteolipids were extracted from total mouse brains and separated from lipids and fats by SEC. Figure 4A shows the separation of proteins from the large quantity of fat and lipid in the chloroform extract. Larger lipids such as cardiolipin and triacylglycerides elute a little earlier than general phospholipids, and so the window for lipid-depleted proteins is narrower. Nevertheless, the proteins can be largely purified for further analysis. Figure 4B shows the mass spectrum and the deconvoluted molecular mass profile, demonstrating some recognizable, abundant proteolipids and other proteins exhibiting this behavior with lower signal to noise ratios. The protein of mass 7653 Da is tentatively identified as the c-subunit of mitochondrial ATP synthase FO with a calculated average mass of 7650 Da including trimethylation of Lys 104 (Q9CR84, residues 62–136). The protein of mass 15,723 Da is tentatively identified as the equivalent subunit of the vacuolar ATPase VatL (P63082) with its initiating Met removed and acetylation of the *N*-terminus (calculated 15719 Da) (see CTDP Proteoform Atlas PFR8641). Clearly, high-resolution top–down mass spectrometry is required to prove the identity of these proteoforms.

## 4. Discussion

As Yergey pointed out [25], it is important to remember that proteomics is a discipline within the broader definition of analytical chemistry and, therefore, a good proteomics study should recognize the relevant overarching principles that center upon accuracy of identification, and quantification, of a chemical entity. For a membrane protein, we want to know how it associates with the biological membrane—whether it is integral or peripheral, whether it binds other proteins, lipids and other small molecules and whether, and how, these are related to that protein’s function. When we can answer these questions for an entire biological membrane, we may claim we have defined the *proteolipidome*. At some point in the future, such information is necessary for a molecular dynamics simulation of the whole membrane and allows the testing of hypotheses in silico before performing biological engineering experiments [26].

Any physical separation that yields a simplified protein preparation has the potential to be useful for proteomics and, in this manuscript, we consider a potential role for partition into an organic solvent to define a hydrophobic subproteome. That some proteins, traditionally the proteolipids, partition into a chloroform-enriched phase of a chloroform/methanol/water phase separation is superficially attractive, especially once proteins have been separated from lipids, in this case, by size-exclusion chromatography. As we show here, these proteins can be separated by reverse-phase chromatography and characterized in complete molecular detail by top–down high-resolution mass spectrometry (Figure 3). The advantage of gaining access to, and enriching the most hydrophobic proteins of the bilayer must be weighed against the disadvantage that treatment with chloroform is highly effective at breaking non-covalent associations leading to loss of information on binding of lipids and other molecules that might modulate function. Though we have been generating a proteolipid fraction for over ten years (first published in 2007 for spinach AtpH), we have never systematically studied the conditions for optimal reproducibility of this preparation relying on the published protocol of Wessel and Flugge [19]; amongst the most important factors, conditions to be optimized include the protein/lipid ratio of the starting material and the relative amount of this fraction compared to chloroform and other solvents, salt concentration, the pH and temperature used for the experiment, length of time/speed of vortex mixing and centrifugation parameters. Such an analysis will be necessary to validate a quantitative proteomics method [25].

The transmembrane alpha helix presents challenging properties that make contemporary trypsin digest-based proteomics inefficient. These helices nearly always lack basic residues for cleavage specificity and are highly enriched in residues with hydrophobic side chains. For larger proteins, this may be of little significance because peptides from loop regions provide identification and quantification. However, studies using chymotrypsin or where cyanogen bromide was used to cleave at Met residues within transmembrane helices, making for shorter, more recoverable peptides as well as other strategies such as elevated column temperatures, have expanded membrane proteome coverage [4,5]. An alternative to cleaving proteins for proteomics is to work with the intact protein using top–down proteomics mass spectrometry [9,18,27,28]. Transmembrane domains are relatively easy to dissociate in the gas phase, provided some heat is provided to melt the otherwise stable alpha helices [15]. Here, we provide the example of the PsaI protein that yields useful sequence tags using both collisionally activated dissociation (CAD) and activated-ion electron-capture dissociation (aiECD) (Figure 3). CAD is a thermal process whereby collisions with inert gas eventually deposit sufficient energy for CAD to occur at the peptide bond. ECD uses energetic electrons to initiate a different dissociation mechanism that primarily cleaves the amide alpha–carbon bond; efficiency is improved when the target cation is thermally activated using infra-red radiation to melt alpha helices (aiECD). Notably, both methods of generation of sequence-dependent information produce long sequence tags that allow definitive proteoform identification despite the limited repertoire of residues within transmembrane domains (Figure 3). The elevated hydrophobicity of both PsaI and AtpH (previously characterized by top–down MS; [15]) is emphasized by Gravy scores of 1.324 and 1.035, respectively (the majority of thylakoid membrane proteins have Gravy scores of less than 0 [23]). In Figure 4, we provide evidence that this approach is applicable to mammalian brain, with two highly abundant proteins with elevated hydrophobicity (Mus mitochondrial ATPase c-subunit and the vacuolar ATPase equivalent VatL have Gravy scores of 1.143 and 1.070, respectively) tentatively identified by intact mass analysis.

New designs of mass spectrometers better suited to native, intact protein work are making novel strategies in structural proteomics possible. Improved designs of interface regions enable electrospray ionization of native, oligomeric complexes with bound co-factors [29]. Subsequent excitation allows displacement of bound small molecules such as lipids and ligands as well as some dissociation analysis of subunits by the top–down approach [30]. These rapidly developing strategies in native mass spectrometry have the potential to revolutionize study of the *proteolipidome.* However, care must be taken in interpretation of complex spectra and, given such controversies, it is perhaps advisable to perform detailed subunit analyses in parallel with native mass spectrometry experiments [31].

## Figures and Tables

**Figure 1 proteomes-08-00005-f001:**
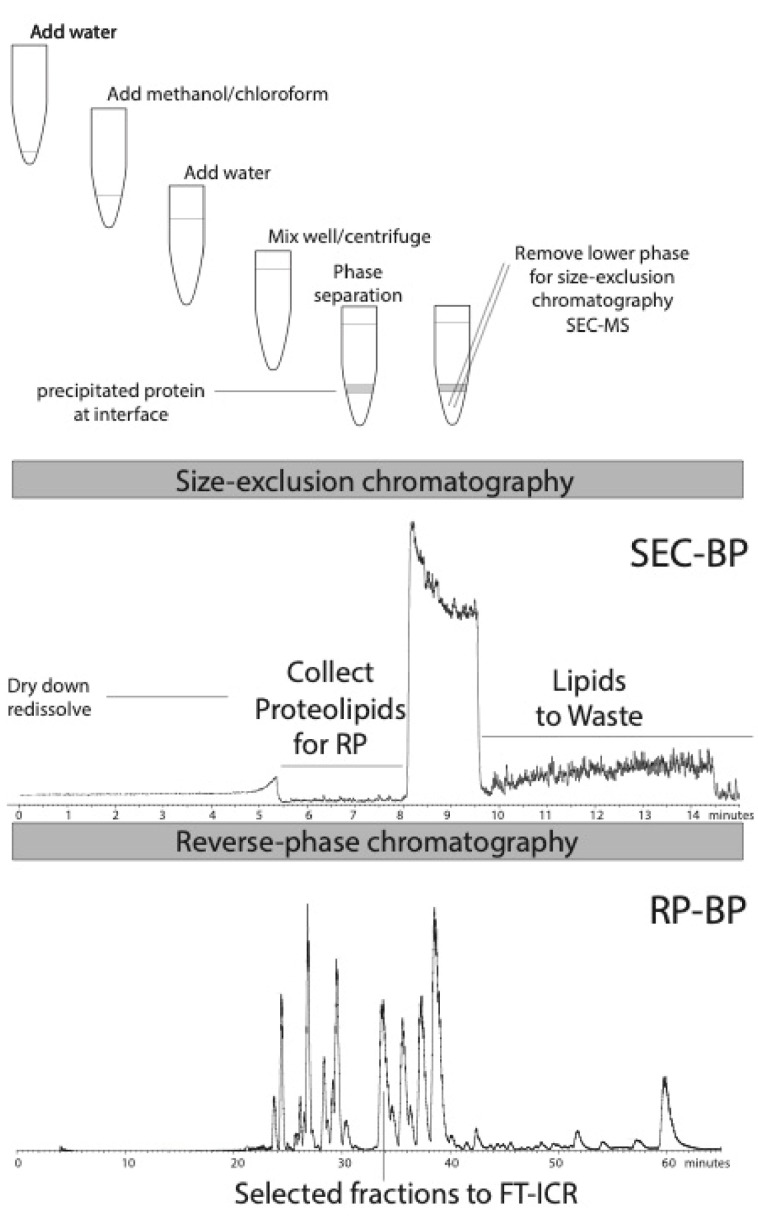
Schematic diagram of extraction of spinach thylakoid membrane proteolipids followed by size-exclusion and reverse-phase chromatography. Chloroform extraction selectively captures lipids and proteolipids in the lower chloroform-enriched phase. This lower phase is then injected to size-exclusion chromatography in order to separate proteolipids from lipids. Proteolipid-containing fractions are dried, redissolved in 70% formic acid and run on reverse-phase (RP) chromatography with low-resolution electrospray ionization mass spectrometry and fraction collection (LC–MS+). Selected fractions are then subject to top–down Fourier-transform ion cyclotron resonance (FT-ICR) MS offline using nano-electrospray ionization. Representative base peak (BP) chromatograms for size-exclusion chromatography (SEC) and RP are shown. The SEC-BP chromatogram shows dropout of signal as the proteolipid peak is collected and again as the bulk of the lipid peak is diverted to waste.

**Figure 2 proteomes-08-00005-f002:**
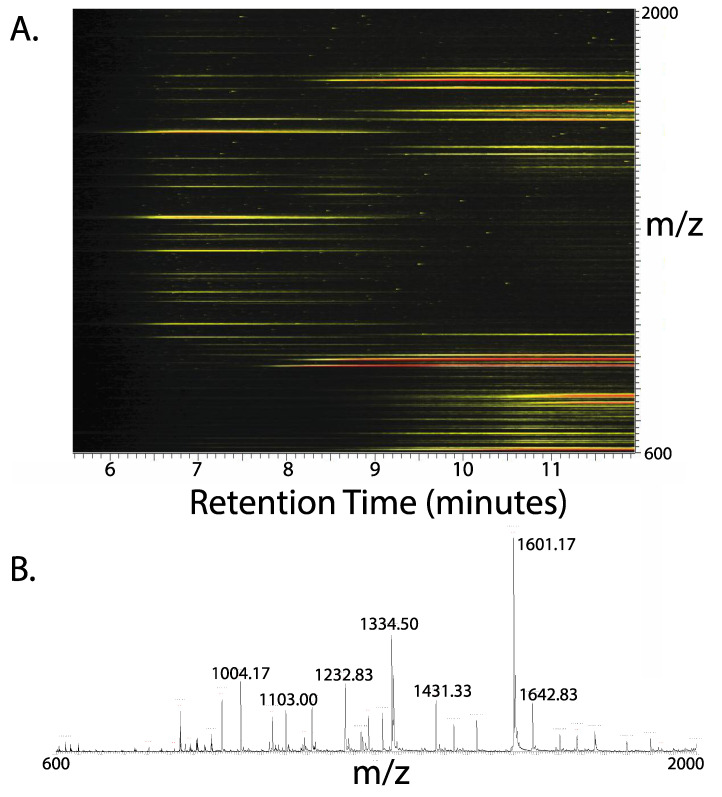
Extraction of spinach thylakoid membrane proteolipids. (**A**) A 3D ion map view of the size-exclusion separation of chloroform-extractable lipids and proteolipids. (**B**) Mass spectrum of multiply charged proteolipid ions averaged from 6.4 to 7.8 min. A deconvolution was not included because several small proteolipids were represented by a single charge only confounding the software.

**Figure 3 proteomes-08-00005-f003:**
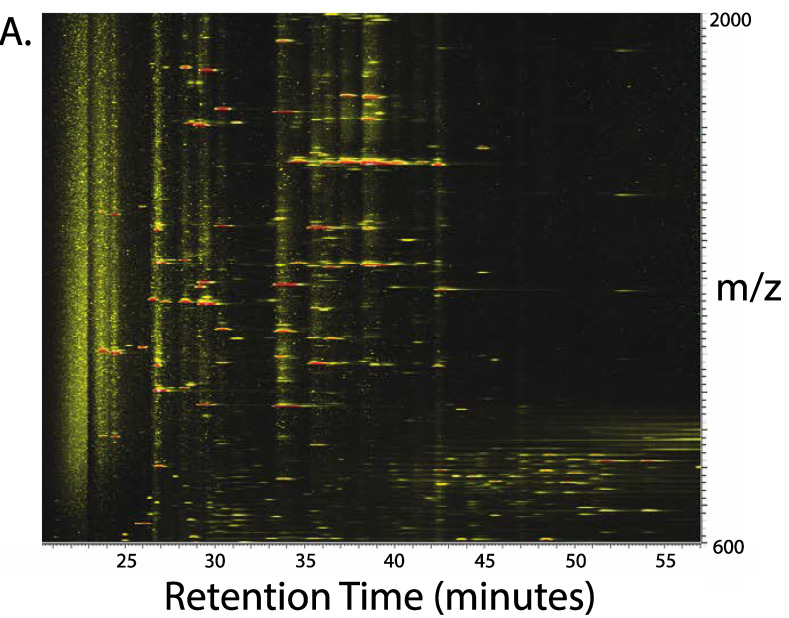
Top–down mass spectrometry of thylakoid proteolipids after SEC purification, resolubilization in formic acid and reverse-phase chromatography. (**A**) A 3D ion map of the chromatographic profile of the reverse-phase separation of spinach thylakoid proteolipids after removal of most lipids by size-exclusion chromatography—as described in Figure 1. Selected fractions collected during the reverse-phase separation were subjected to static nanospray tandem mass spectrometry. (**B**) Top–down mass spectrometry of an unknown intact mass tag (3696.9 Da) eluting at 29.5 min was selected for identification. Collisionally activated dissociation (CAD) was performed on the triply charged precursor, *m*/*z* 1233.3 Da (monoisotopic mass 3694.038 Da). The ion isolation of the precursor is shown in the inset. The CAD spectrum was processed using Prosight 4.1 software to generate sequence tags, and in turn identify the protein as PsaI. The PsaI sequence is shown with the transmembrane helix shaded. (**C**) Activated-ion electron capture dissociation (aiECD) was used to dissociate the triply charged precursor at *m*/*z* 1233.3 Da. The aiECD spectrum was processed using Prosight 4.1 software to generate sequence tags, and in turn identify the protein as PsaI.

**Figure 4 proteomes-08-00005-f004:**
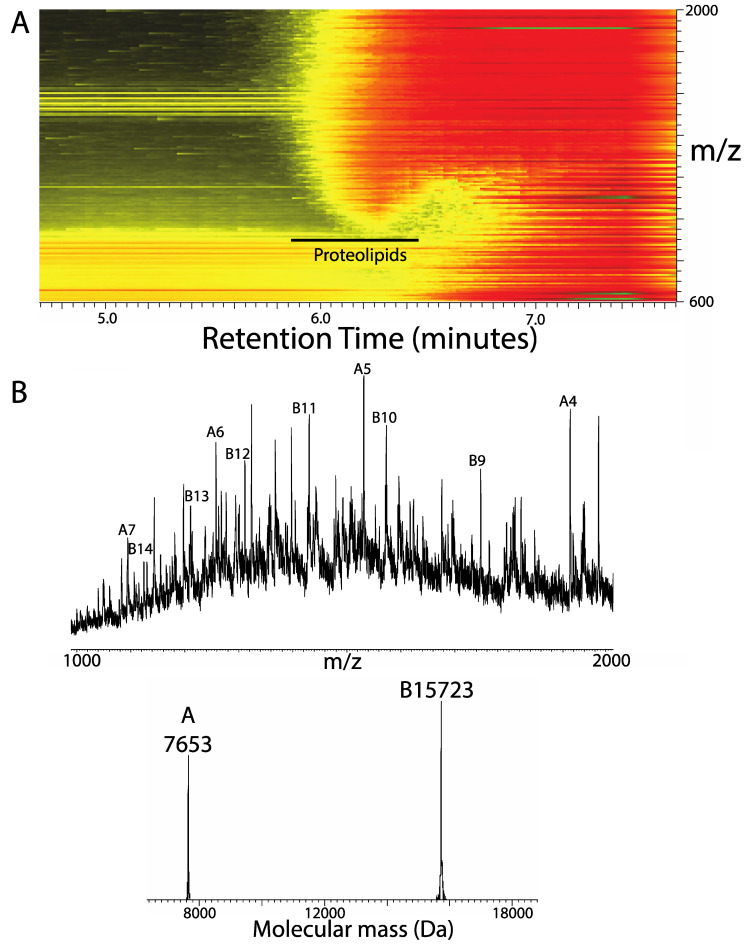
Extraction of mouse brain proteolipids. (**A**) A 3D ion map view of the size-exclusion separation of chloroform-extractable lipids and proteolipids. (**B**) Mass spectrum (upper) and Molecular mass profile after deconvolution using MagTran software (lower panel) of prominent proteolipids—as labeled. Ions that correspond to the deconvoluted masses are labeled A and B according to their origin, with the number corresponding to the charge state.

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
