# Peer review of "Targeting a Subset of the Membrane Proteome for Top–Down Mass Spectrometry: Introducing the Proteolipidome"

_proteomes, 2020, doi:10.3390/proteomes8010005_

Round 1

Reviewer 1 Report

This well written manuscript reports a critical refinements in the technology for analysis of membrane proteins that has been under development in Dr. Whitelegge's lab for more than 20 years. These improvements will be of interest to the fairly large community of scientists who study membrane proteins. Based on experimental success, the author also proposes to define the proteolipidome as a new subproteome. I appreciate his candid reporting about the efficiency of SEC. I am enthusiastic about this manuscript, with one exception that I feel must be corrected or clarified. I also make some minor suggestions intended to be helpful.

Minor:

in the abstract please insert "spinach' in front of "thylakoid membrane." This will further emphasize the importance of the experiment with mouse brain. In the Legend to Figure 2 please insert thylakoid. Please indicate in the introduction whether covalently bound lipoproteins are expected in the chloroform soluble sample and thus the analysis. I suggest that the Methods are not all sufficiently described for me to reproduce the experiment. Perhaps the author could insert references to his relevant earlier publications into the Methods section? In particular, could we learn more about CAD and aiECD conditions? What was the target gas for CAD?  What was the reaction time for aiECD?  Please indicate in the Results what is the source of the protein database for S. oleracea. Is this from UniProt? If it is a custom database is it available to the public? Perhaps this would belong in the Methods section.  In the penultimate paragraph of the manuscript the author indicates that 'heat' should be applied for top-down analysis of alpha helices. Would he please indicate here and in the methods section how that was done in his experiment? Also in the penultimate paragraph the last sentence might be improved by saying that Figure 3 "indicates" or "suggests" in place of "demonstrate." I recommend that the Legend to Figure 3 be clarified. Please tell us if the "mass spectrum" in B is from a direct insertion sample or from what time range in the chromatogram. Please indicate how the "molecular weight profile" is derived. Has the entire "mass spectrum" been deconvoluted? In the discussion preceding Figure 3 you say that many other proteins are revealed in Figure 3B. Please tell us why these do not show up in the decunvoluted molecular weight profile shown.              Figure 3A is very impressive. 

Major concern. In the first paragraph of the Results section you write that the 3-D contour mass chromatogram in Figure 1A shows that proteins elute in 6-8 minutes while lipids elute in >8 min. Figure 1A is labeled Ion map view. Is this the same? The time axis of the ion map view only runs to 6.0 min. Thus it cannot provide information in the 6-8 minute range. Please rectify. Also, at what time or time range is the spectrum in Figure 1B recorded? 

Reviewer 2 Report

In the paper entitled “Targeting a Subset of the Membrane Proteome for Top-down Mass Spectrometry; Introducing the Proteolipidome” by J. Whitelegge (Proteomes-712067), a chloroform/methanol/water phase partition method was applied to spinach and mice brain tissues. The recovered lipid fraction (bottom phase) was subjected to size exclusion chromatography (SEC) fractionation to isolate proteolipid fractions. These fractions were further separated by reverse-phase liquid chromatography (RPLC) followed by top-down mass spectrometry (MS) analysis. One proteolipid (Psa1 protein) was identified using tandem MS from spinach extract. Two other proteolipids are tentatively identified in the brain extract based on accurate masses alone.

This method paper is interesting and relevant to scientists studying cell membranes however it has many flaws both major and minor as detailed below in a chronological fashion.

The most crucial omission to this study is the lack of biological and technical replicates. No method can be validated without demonstrating its reproducibility! Furthermore, replicates are needed to demonstrate that this method is quantitative. This aspect must absolutely be addressed.

Moreover, how does the author know that proteolipids elute along the chromatogram where he says they elute without identification results? They could be entirely different compounds. There is no evidence of this and must be rectified.

As a method paper, all the technical information needed to repeat experiment should be supplied and it’s not the case, as detailed below. But this should be easily addressed.

Abstract

L16-17: replace “upward” and “downward” with > and <

Introduction:

Please define proteolipids and explain how they differ from lipoproteins to avoid confusion.

L47: whilst trypsin is the gold standard for protein digestion, it is not suited to membrane proteins. Chymotrypsin is a suitable alternative. Please specify this with appropriate citations.

L48-54: these 3 sentences contain the verb “yield” 4 times! Replace with synonyms.

L52-53: this sentence “Intact membrane proteins have yielded to chromatography under specialized conditions empowering top-down mass spectrometry that includes the transmembrane domains” is weirdly put. Please rephrase.

L59-62: citations 9 and 10 referring to proteolipids and ATPase are very old (1953 and 1976). Please also include more recent relevant literature.

L62-64: ATPase was not the 1st protein analysed by top-down proteomics. Please rephrase.

Materials and Methods

This section is greatly incomplete which is unacceptable for a method paper. Furthermore, it lacks clarity, hard to follow and therefore needs a schematic diagram illustrating step by step the workflow. It must be clear whether SEC fractions were collected, and then some were disregarded (lipids) and others (with proteolipids) were pooled for subsequent RP-HPLC (if that’s what happened; I was very confused while reading it). Explicitly outline RPLC fraction collection and MS analyses (online vs offline). Please add this diagram as Figure 1.

Please complete missing details as listed below.

L73: which organ/tissue of spinach was used? How much plant tissue was processed? How was the plant tissue pulverised? How were thylakoid membranes prepared?

L75: indicate molarity and pH of PBS.

L76: indicate temperature of centrifugation step.

L78: what exactly are the samples which underwent phase partition?

L81: typo for “vigoroursly”.

L81: indicate temperature of centrifugation step.

L82: specify Hamilton syringe volume.

L84: what LC system(s) was (were) used?

L85: v/v/v (there are 3 solutions mixed together)

L86: describe SEC fully. It’s a method paper, please be thorough (flow rate, temperature, mobile phases, gradient, duration…)

L87: describe column equilibration

L88: indicate make/model of mass analyser along with all the parameters.

L89: fraction volume? How many fractions collected?

L90: what exactly are the samples which underwent RPLC?

L91: mobile phases are not buffers but mere solutions! Please rectify.

L91: describe RPLC fully (sample volume, flow rate, temperature…)

L96-99: online vs. offline MS. Please clarify when one was used and the other. This must be explicit in the schematic diagram too. Indicate all the parameters used for both MS instruments. Again, be thorough please.

L100: explain “static” nESI. This is an usual term.

L105: indicate MagTran and Prosight parameters. What was the database searched? Give lots of details (format, number of entries, version, realease date…) and possibly a web link to it.

Results:

L114: “for a maximum of 24-hour”. Why? If this is explained later on L132-134, move the sentence here.

L115-116: “Aliquots (100 μL) were injected onto a 114 size-exclusion column (SW2000; Tosoh Biosciences) previously equilibrated in chloroform/methanol/1% aqueous formic acid (4/4/1; v/v) at 40 ˚C.” This has already been specified in the methods section. Remove.

L116: “Proteins that precipitate”. Why do you bring this up? It that what you observed in your experiment? It comes out of nowhere. Please elaborate.

L120-121: “maximal protein purification is achieved while yield is limited”. this seems contradictory to me, please rephrase.

L122: typo, change “fraction” to “fractions”.

L125: the elution profile mentioned here (6-8 min and >8 min) cannot be checked on Fig 1 which only displays 0-6 min elution. Please rectify Fig 1.

Fig 1: add LC chromatogram, intensity scale of ion map, y-axis and m/z range of spectrum. Show Xcalibur header names as well (in QualBrowser, a chromatogram/spectrum/map header contains a lot of valuable information, please show). Indicate RT of spectrum.

In fig 1, lots of peaks coelute for more than 1 min. Why was the 1 min window collection chosen? Were some of the fractions pooled later on?

Please supply the following supplemental data: the whole map (no zoom in) of spinach and brain samples for both SEC and RPLC MS analyses (therefore 4 ion maps. Feel free to include zoom-in sections as well).

Could you also provide as a table the list of all the deconvoluted masses and their abundances?

L141: why did you choose this feature (3696.9 Da)? Explain the rationale here and also populate the Methods accordingly. Show deconvolution data of this protein.

L145: indicate Psa1 sequence coverage overall (considering both CAD and aiECD). Could you please discuss how CAD and aiECD operate and why they produce such complementary results? This will nicely highlight the need of employing various MS/MS fragmentation modes in order to achieve greater sequence coverage.

L149: supply the evidence that indeed “This protein retained its initiating formyl-methionine residue accounting for the extra 28 Da mass”. Formylation could arise from the formation of FA adducts during the ionisation. You should mention this artefact as well. A proof of artefactual formylation would be if all the identified proteins systematically display this 28 Da mass shift.

Did you see any other PTMs of this protein? If so, please include those results in your paper.

Fig 2: add LC chromatogram, intensity scale of ion map, y-axis and m/z range of spectrum. Which SEC fraction was used to produce Fig 2A? Show Xcalibur headers as well (in QualBrowser, a chromatogram/spectrum/map header contains a lot of valuable information, please show). Add MS1 spectrum as well with the deconvoluted data as an inset. Add transmembrane domain to the Psa sequence.

Legend Fig 2: remove “– as in Figure 1”, this is not what happened. Indicate RT of spectrum.

L174: supply numbers for this narrower window.

L176: “recognizable”. How so?

L177-181: using MS1 accurate mass only, the author suggests some proteolipid IDs. Since these masses are small, these compounds could easily be analysed by HRAM MS/MS as previously done on Psa1. Please perform this top-down experiment. Otherwise, there is no point forthcoming potential IDs. The list of candidates is simply too great based on mass only!

Fig 3: again add LC chromatogram, intensity scale of ion map, y-axis and m/z range of spectrum, headers. In legend indicate “Molecular weight profile following deconvolution using MagTran”

In fig 3, how do you know proteolipids elute from 7.2-7.9 min? What are the other compounds? Could you point to the lipids and the proteins? Give the evidence.

Discussion:

L210: as stated before, chymotrypsin is more efficient at digesting membrane proteins than trypsin. Please include with references.

Round 2

Reviewer 2 Report

In the revised version, all of my minor comments were appropriately addressed. The methods are more complete and include a diagram that helps in understanding the workflow (Fig 1).

There are several typographical/grammatical mistakes to correct, which warrant another careful reading of the manuscript.

My previous two major comments pertained to 1/ to the absence of replicates, and 2/ to the lack of identification evidence of the elution of proteolipids from 5-8 min SEC. In their cover letter the author justifies these shortcomings on page 2 but I couldn’t find such disclaimers in the actual manuscript.

Typically, where is the following sentence “Though we have been generating a proteolipid fraction for over ten years (first published in 2007 for spinach AtpH) we have never systematically studied the conditions for optimal reproducibility of this preparation relying on the published protocol of Wessel and Flugge [Wessel, 1984]; conditions to be optimized include protein/lipid ratio of starting material and the relative amount of this fraction to chloroform and other solvents, salt concentration, the pH and temperature used for the experiment, length of time/speed of vortex mixing and centrifugation parameters amongst the most important factors. Such a analysis will be necessary to validate a quantitative proteomics method.” (see cover letter, page 2)?

Likewise, where is the sentence “It is assumed that all these multiply charged ions result from proteolipids but until we have formally proven this to be so we have to entertain the possibility that some other biological macromolecule might be included in this fraction.” (see cover letter, page 2)?
